# Interactions with bacteria shape diatom adaptation to carbon concentration changes

Chenjie Li, Wenxiu Yin, Yufang Pan & Hanhua Hu ✉

Diatoms are key contributors to global primary production, and have developed intricate partnerships with bacteria through long-term co-evolution. Here, we uncover a syntrophic relationship between the model obligate photoautotroph diatom *Phaeodactylum tricornutum* and the rod-shaped bacterium *Loktanella vestfoldensis*, which enables the diatom to indirectly utilize glucose. To be specific, growth of the diatom depends on the support of *L. vestfoldensis* for the supply of necessary carbon source when glucose serves as the sole carbon source, while *L. vestfoldensis* shows dependence on *P. tricornutum* when $CO_2$ is the sole carbon source. Reanalysis of *Tara* Oceans metagenomic data shows frequent co-occurrence of *Loktanella* with diatoms including *Chaetoceros* and *Thalassiosira*, indicating the ecological relevance of this partnership. Co-culture with *L. vestfoldensis* supports robust growth of *Chaetoceros muelleri* and *Thalassiosira pseudonana* in the presence of glucose as the sole carbon source. Transcriptomic and metabolomic analyses reveal that *P. tricornutum* maintains a photoautotrophic metabolism in co-culture, as indicated by the up-regulation of genes involved in inorganic carbon concentration and photosynthesis, while the co-cultured bacterium likely supplies $CO_2$ and growth-stimulating metabolites such as indole-3-acetic acid. Our findings demonstrate that bacterial-algal interactions may shape diatom adaptation to carbon changes and contribute to marine carbon cycling.

Diatoms, one of the most abundant and diverse groups within the phytoplankton community, constitute crucial primary producers in marine ecosystems and account for approximately 20% of global primary productivity[1,2]. They play a significant role in global biogeochemical cycles, particularly in the carbon, nitrogen, sulfur, and silicon cycle[3]. Due to siliceous cell walls, diatoms are prone to sedimentation and form a net carbon flux towards the lower layer of seawater, and are considered the main participants in the carbon pump of marine organisms[4].

In surface seawater, despite the high concentration of dissolved inorganic carbon, which is predominantly $HCO_3^-$ and relatively less $CO_2$, the depletion of $CO_2$ compounded by intense phytoplankton photosynthesis poses constraints to the growth of diatoms. Concurrently, due to the low affinity of ribulose-1,5-bisphosphate carboxylase/oxygenase (Rubisco) for $CO_2$, nearly all marine diatoms have

evolved biophysical and potential biochemical inorganic carbon concentrating mechanisms (CCMs) to maintain higher $CO_2$ concentrations around Rubisco and ensure the efficient photosynthesis[5]. The biophysical mechanism involves active $HCO_3^-$ transport into the cytoplasm via solute carrier family 4 (SLC4), with cytoplasmic and chloroplast carbonic anhydrases (CAs) mediating the interconversion between $HCO_3^-$ and $CO_2$[6,7]. Biochemical pathways may involve phosphoenolpyruvate carboxylase (PEPC)-mediated $C_4$ metabolism, which is still a finding for debate though[5]. In addition, many marine planktonic algae, including diatoms, exhibit mixotrophic capabilities, combining the phototrophic and heterotrophic modes[8]. Mixotrophy provides a crucial advantage for marine plankton and enhances their resilience in marine food webs, while contributing a ~35% increase in the carbon flux to higher trophic levels[9]. For example, genome-scale modeling of *Cylindrotheca closterium* predicts that mixotrophic

Key Laboratory of Algal Biology, Institute of Hydrobiology, Chinese Academy of Sciences, Wuhan, China. ✉e-mail: hanhuahu@ihb.ac.cn

growth (with both inorganic and organic carbon sources) driven possibly by algal-bacterial interactions predominates, which accounts for 71%[10].

Diatom-bacterial interactions encompass nutrient exchange, chemical signaling, and community regulation. Bacteria closely associated with diatoms include genus *Sulfitobacter*, *Roseobacter*, *Alteromonas*, and *Flavobacterium*[11]. Diatoms utilize nitrogen and vitamins provided by bacteria, and in turn, they provide bacteria with essential organic matters[11–13]. However, these relationships exhibit antagonistic dimensions, with diatoms producing antimicrobial compounds and bacteria deploying algicidal strategies[14,15]. To sustain photosynthetic carbon fixation, the model diatom *Phaeodactylum tricornutum* employs biophysical CCMs through SLC4 transporters and 11 distinct CAs[6,7], and partially through biochemical $HCO_3^-$ fixation catalyzed by the mitochondrial PEPC[16]. This species also demonstrates mixotrophic capacity through exogenous organic carbon uptake, and the biomass and lipid production are enhanced[17]. In addition, growth and pigment accumulation of *P. tricornutum* can be promoted by bacterial interactions[18], however, the mechanism of bacterial-algal interactions in mixotrophy is poorly studied.

In this study, we identify a *P. tricornutum* strain CCMM 2004 (PtCr), capable of utilizing exogenous glucose indirectly, from 12 geographically distinct isolates. Meanwhile, the Roseobacter *Loktanella vestfoldensis* is isolated from the PtCr culture, and it is the

bacterium that facilitates the growth of *P. tricornutum* when glucose is used as the sole carbon source. Transcriptomic and metabolomic analyses reveal the key role of the bacterium on diatom growth under inorganic carbon-limited conditions, and the dominant co-occurrence of *Loktanella* and marine diatoms is also confirmed by our reanalysis of *Tara* Oceans macrogenomic data. Our study reveals important insights into the interaction between bacteria and diatoms under different availability of inorganic and organic carbon, and provides a basis for explaining the adaptation of marine diatoms to inorganic carbon-limited environments.

## Results

### Growth of 12 *P. tricornutum* strains with different carbon sources

It is well-established that the diatom *P. tricornutum* is an obligate photoautotroph, incapable of heterotrophic growth. It is reasonable that no growth was observed in the media without supplemented carbon sources (CL: carbon limitation) for all 12 *P. tricornutum* strains, and in the media with glucose as the sole carbon source (GC) for strains Pt1-Pt10 and UTEX640 (Fig. 1a and Supplementary Fig. 1a). Accordingly, no obvious glucose consumption was detected in the cultures of strains Pt1-Pt10 and UTEX640 (Fig. 1a and Supplementary Fig. 1b). On the contrary, all strains exhibited substantial growth with atmospheric $CO_2$ as the sole carbon source (AC), reaching the final

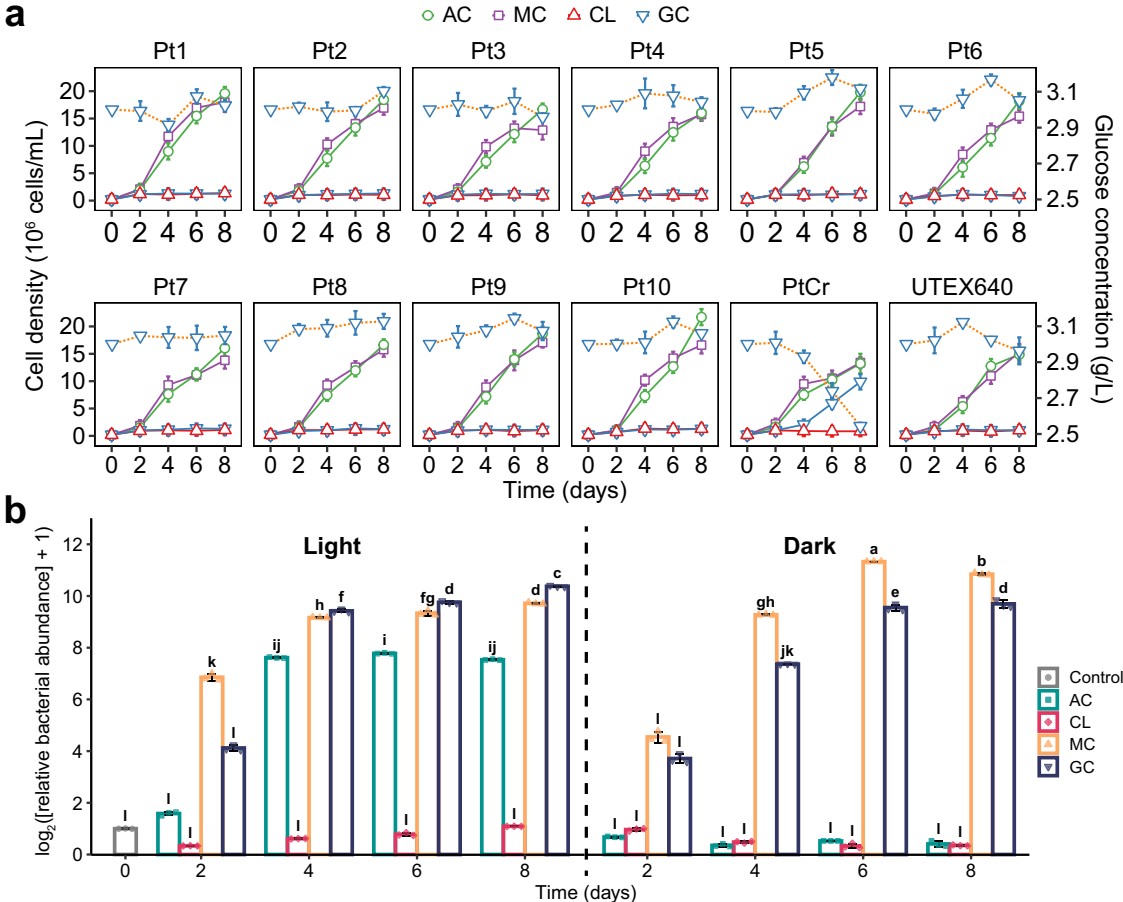

**Fig. 1 | Growth of *P. tricornutum* strains and abundance of co-cultured *L. vestfoldensis* under varying carbon sources and light conditions.** Cell density and glucose concentration (dot orange line) of 12 strains of *P. tricornutum* in cultures with light (**a**), and relative abundance of *L. vestfoldensis* co-cultured with strain PtCr incubated with light and in dark, respectively (**b**). AC, using atmospheric $CO_2$ as the sole carbon source; MC, using mixed carbon sources (atmospheric $CO_2$ and 3 g/L glucose); CL, carbon limitation; GC, using 3 g/L glucose as the sole carbon source; Control, initial bacterial abundance. The line and bar plots represent mean ± standard error ($n = 3$ biological replicates). Different letters above the bars indicate statistically significant differences by two-tailed Fisher's Least Significant Difference (LSD) test (66 degrees of freedom, $p < 0.05$, Benjamini-Hochberg correction). Source data are provided as a Source data file.

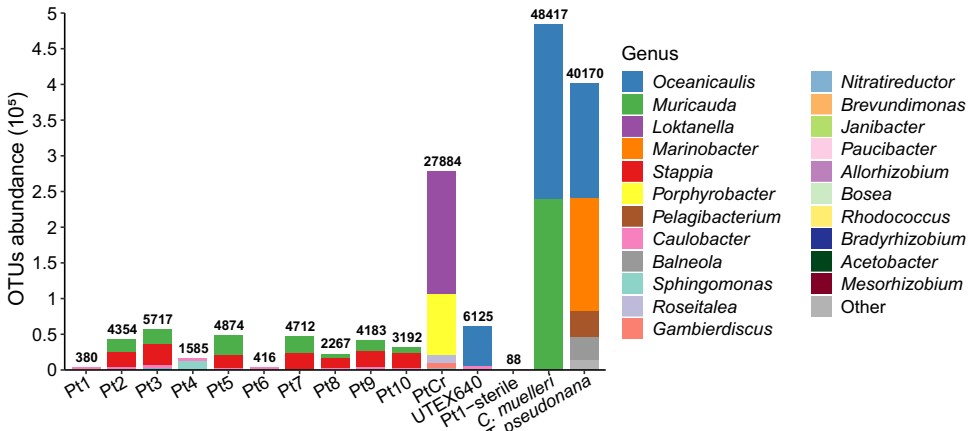

**Fig. 2 | Estimation of OTUs abundance in associated bacteria of 13 *P. tricornutum* strains together with *C. muelleri* and *T. pseudonana*.** The four bacterial communities with the highest OTUs abundance per sample at the genus level were showed. Pt1-sterile, Pt1 with intensified antibiotic treatment. Source data are provided as a Source data file.

densities of $1.3–2.2 \times 10^7$ cells/mL ($1.7–2.2 \times 10^7$ cells/mL with silicate in the media) after 8 days. An extra addition of glucose to the media under AC conditions (MC: mixed carbon) did not improve the algal growth.

As the only exception, strain PtCr (receiving standard antibiotic treatment) seemed to be a "glucose-utilizing" strain which could grow with glucose as the sole carbon source (GC) and achieved a final density of $1.0–1.1 \times 10^7$ cells/mL on day 8 with the glucose depletion of 456–538 mg/L (Fig. 1a and Supplementary Fig. 1a, b). pH values (7.1–8.4) in the culture of PtCr showed a similar trend with those in other Pt cultures (Supplementary Fig. 1c), and its inorganic carbon concentration was comparable with that of Pt1 under GC condition (Supplementary Fig. 2).

To further elucidate the seeming "glucose-utilization" of PtCr, its growth under darkness was observed and then compared with that of Pt1 (receiving standard antibiotic treatment) and Pt1-sterile (receiving intensive antibiotic treatment). In an apparent contrast with the illumination condition, PtCr could not grow under darkness with atmospheric $CO_2$ and glucose as carbon sources (MC-Dark) or with glucose as the sole carbon source (GC-Dark). Likewise, no growth of Pt1 and Pt1-sterile was detected under darkness (Supplementary Fig. 3a). However, apparent glucose consumption (271–693 mg/L in PtCr and 296–321 mg/L in Pt1 after 8 days) was detected in the cultures of both strains, Pt1 and PtCr, under darkness (Supplementary Fig. 3b), but not in strain Pt1-sterile. These results suggested that strains PtCr and Pt1 could not grow heterotrophically on glucose, and the decrease of glucose in the cultures should be attributed to epiphytic bacteria, which cannot be eradicated by conventional antibiotics.

### Identification of bacterial community and growth of epiphytic bacteria

To identify the bacterial community of 13 *P. tricornutum* strains together with *Chaetoceros muelleri* and *Thalassiosira pseudonana*, 16S regions (V3 + V4) were sequenced, which revealed a total of 1.09 million clean reads (Supplementary Table 1). Abundance of operational taxonomic units (OTUs) in *C. muelleri* (48417) was the highest, while the lowest was observed in Pt1-sterile (88). Eleven classes, 20 orders, 29 families, and 42 genera were identified, with Alphaproteobacteria and Bacteroidia contributing the major OTUs abundance. Notably, *Janibacter* sp. only appeared in Pt1, and *L. vestfoldensis* was only present in PtCr (Fig. 2, Supplementary Figs. 4, 5 and Supplementary Data 1).

The epiphytic bacterium *L. vestfoldensis* (class Alphaproteobacteria, Rhodobacter group) was isolated from the PtCr cultures (Supplementary Table 2), and its growth in the PtCr cultures was also

investigated. The abundance of *L. vestfoldensis* increased over time in the media with added glucose (GC and MC conditions) under both light and dark conditions, and higher bacterial abundance was reached from day 6–8 under MC condition with darkness and under GC condition with light (Fig. 1b). When atmospheric $CO_2$ was used as the sole carbon source (AC), bacterial abundance increased significantly only under light. No significant increase of bacterial abundance was detected in the media without supplemented carbon sources (CL) under light or darkness (Fig. 1b). We also isolated an epiphytic bacterium from the Pt1 cultures and identified it as *Janibacter anophelis* within the family Intrasporangiaceae (Supplementary Table 2).

### Growth of 12 *P. tricornutum* strains supplemented with *L. vestfoldensis*

To assess bacterial-mediated glucose utilization, we introduced *L. vestfoldensis* to all the 12 *P. tricornutum* strains with glucose as the sole carbon source (GC) under illumination. An extra addition of *L. vestfoldensis* to the PtCr culture had little effect on the algal growth and glucose utilization (Fig. 3). In contrast, the addition of *L. vestfoldensis* supported the growth of the other 11 *P. tricornutum* strains with cell densities ranging from 0.7 to $1.1 \times 10^7$ cells/mL after 8 days (Fig. 3a), a little lower than cell densities of $1.3–2.2 \times 10^7$ cells/mL with atmospheric $CO_2$ as the sole carbon source (Fig. 1a). Accordingly, glucose concentration decreased by 197–494 mg/L in the cultures of the 11 *P. tricornutum* strains after the supplementation of *L. vestfoldensis* (Fig. 3b).

### Co-occurrence of *L. vestfoldensis* and diatoms in global oceans

To determine the long-term coexistence of the isolated bacteria and *P. tricornutum*, we analyzed the global co-occurrence frequency between diatoms and candidate epiphytic bacteria (*Janibacter* and *Loktanella*) using *Tara* Oceans metagenomic datasets. Diatoms were detected at 151 sites, with *Loktanella* co-occurring at 107 sites (frequency = 0.71), compared to only 6 sites for *Janibacter* (frequency = 0.04) (Fig. 4a, b and Supplementary Data 2). This order-of-magnitude disparity in phycosphere association frequencies (0.71 vs. 0.04, Chi-square test, $p < 0.0001$, degrees of freedom: 1, effect size statistic: 0.67, 95% confidence intervals: 59.4% to 74.4%) provides compelling evidence that *L. vestfoldensis*-diatom interactions represent an evolutionarily selected partnership rather than stochastic colonization events. Furthermore, a co-occurrence analysis revealed a clear association of *Loktanella* with the ubiquitous diatom genera *Chaetoceros*, *Thalassiosira*, *Pseudonitzschia*, and *Skeletonema* (Fig. 4c). Addition of *L. vestfoldensis* to the cultures of photoautotrophic *C. muelleri* and *T. pseudonana*

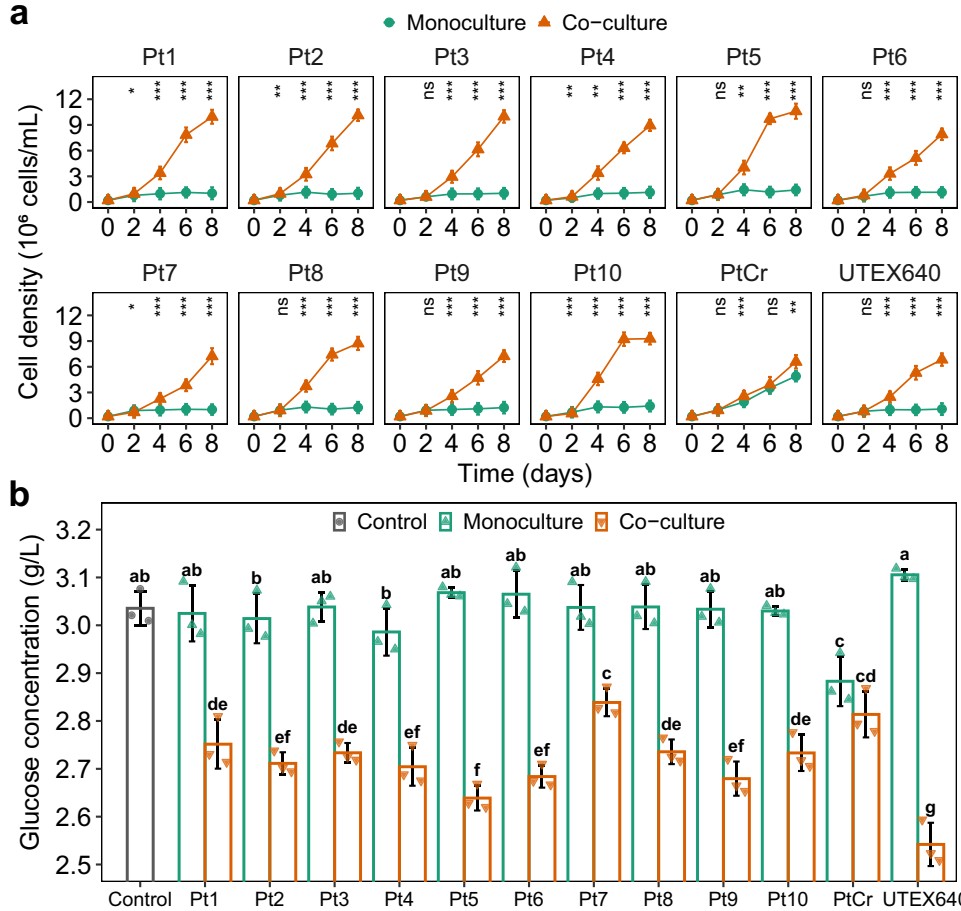

**Fig. 3 | Functional verification of *L. vestfoldensis*. a**, **b** The cell density dynamics and glucose concentrations after 8 day cultivation with the addition of *L. vestfoldensis*. Algal monoculture, monoculture culture of *P. tricornutum*; Co-culture, co-culture of *L. vestfoldensis* and *P. tricornutum*; Control, initial glucose concentration. The line and bar plots represent mean ± standard error (*n* = 3 biological replicates). Asterisk represents statistically significant differences under different conditions for the same cultivation time (two-tailed unpaired *t*-test, 4 degrees of freedom, *$p < 0.05$, **$p < 0.01$, ***$p < 0.001$, ns not significant). Different letters above the bars indicate statistically significant differences by two-tailed Fisher's LSD test (50 degrees of freedom, $p < 0.05$, Benjamini-Hochberg correction). Source data are provided as a Source data file.

(Supplementary Fig. 1a, b) under GC conditions (Fig. 4d, e) supported the growth of the diatoms.

### Transcriptional response of *P. tricornutum* to co-culture with *L. vestfoldensis*

RNA sequencing analysis was performed on *P. tricornutum* monocultures (designated as "algal monoculture") and *P. tricornutum*-*L. vestfoldensis* co-cultures (designated as "co-culture") with glucose as the sole carbon source for 12, 24, 48, and 96 h (Supplementary Table 3 and Supplementary Data 3, 4). Principal component analysis (PCA) and clustering analysis indicated stable transcriptional effects of bacterial-algal interactions after 48 h (Supplementary Fig. 6a, b). Compared to algal monoculture, the number of differentially expressed genes (DEGs) in co-culture increased gradually. Twenty-seven genes, mainly associated with CCM, fatty acid biosynthesis, and carbon metabolism, were consistently up-regulated during 96 h (Supplementary Fig. 6c, d); meanwhile 10 down-regulated genes were mainly related to mitochondrial carrier proteins (Supplementary Fig. 6e and Supplementary Table 4). The number of DEGs with large fold changes increased significantly over time, and the number of strongly up-regulated genes with fold change > 16 and 32 was far more than that of strongly down-regulated ones (Supplementary Fig. 7). KEGG enrichment analysis revealed that up-regulated pathways mainly included photosynthesis, carbon fixation, porphyrin metabolism, nitrogen metabolism and fatty

acid biosynthesis, while down-regulated pathways were mainly fatty acid metabolism and ribosome biogenesis (Supplementary Figs. 8 and 9).

Under inorganic carbon limitation, *P. tricornutum* enhances plastid carbon availability for Rubisco through coordinated regulation of CAs and SLC4 transporters. The diatom employs 11 CA isoforms and six SLC4 homologs, spatially distinct, to achieve this metabolic adaptation. Compared to the algal culture (f/2 enriched medium without NaHCO$_3$ in vent cap culture flasks) before inoculation (0 h), transcript of two genes encoding PtCA1 (pyrenoid-localized) and SLC4-2 (cell membrane-localized) was markedly higher in the medium with glucose as the sole carbon source, indicating the inorganic carbon limitation (Supplementary Fig. 10). Furthermore, the transcript levels of genes encoding CA-III/-VI/-VII localized in the lumen of the chloroplast endoplasmic reticulum were significantly up-regulated in co-culture relative to algal monoculture, and there were no significant changes in the transcript levels of genes encoding CA-I/-II localized in the periplastidal compartment, and CA-VIII/-IX localized in the mitochondria. Transcription of genes encoding θCA localized in the pyrenoid-penetrating thylakoid lumen was significantly up-regulated at 48 and 96 h, whereas that of genes encoding PtCA1 and PtCA2 (pyrenoid-localized) was up-regulated at all 4 time points (Fig. 5). Transcript levels of genes encoding SLC4-2 localized to the cell membrane and SLC4-1 predicted to be localized to intracellular membrane structures

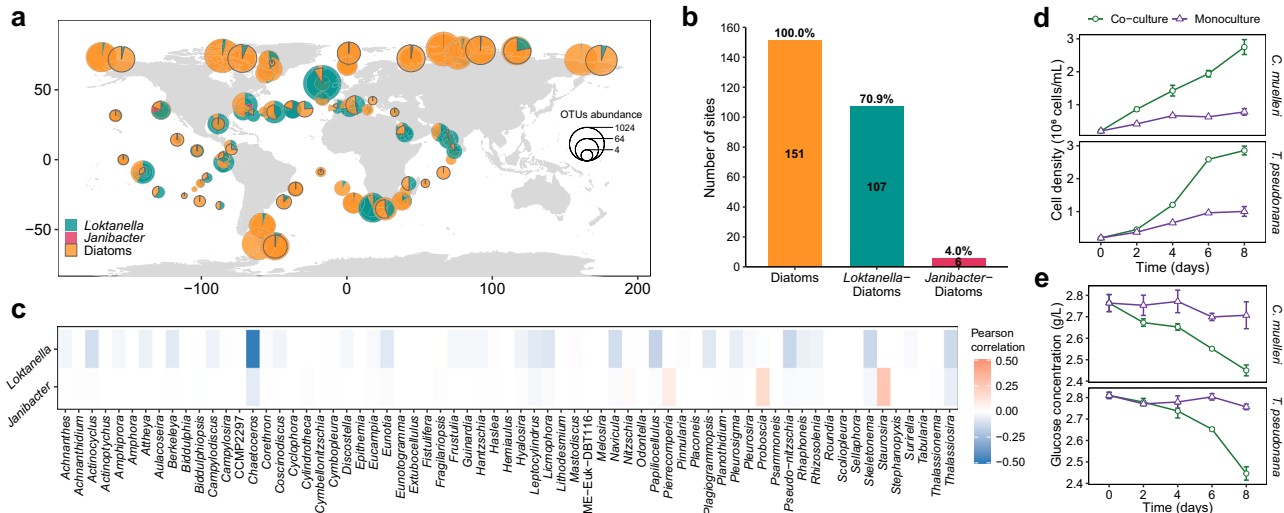

**Fig. 4 | Co-occurrence of *L. vestfoldensis* and diatoms in global oceans and experimental verification in the ubiquitous diatoms.** Distribution (**a**) and overlapping sites (**b**) of diatoms and two bacteria (*Janibacter* and *Loktanella*) according to the metagenomic data from *Tara* Oceans. Geographic visualizations used public-domain map data from the maps package. Sectors of the pie chart represent the proportion of the three groups, and the size represents the abundance of OTUs. Co-occurrence of diatom genera and the two bacteria based on *Tara* Oceans datasets through FlashWeave, FDR < 0.01 (**c**). Cell density dynamics (**d**) and glucose concentrations (**e**) of *C. muelleri* and *T. pseudonana* cultured with glucose as the sole carbon source. The line plots represent mean ± standard error (*n* = 3 biological replicates). Co-culture, co-culture of *L. vestfoldensis* and a diatom; Algal monoculture, monoculture of a diatom. Source data are provided as a Source data file.

were significantly up-regulated at 24 and 48 h. The transcript levels of *SLC4-3/-5/-6-7* exhibited no significant changes, except for that of *SLC4-7* at 96 h (Fig. 5). In addition, transcription of genes encoding the putative $C_4$ pathway enzymes, including two PEPCs, a malate dehydrogenase (MDH1) and a malic enzyme (ME2), was also up-regulated in co-culture relative to algal monoculture (Fig. 5). Subsequently, transcript up-regulation was detected in genes involved in Calvin-Benson cycle, such as *phosphoglycerate kinase, glyceraldehyde-3-phosphate dehydrogenase, triosephosphate isomerase* and *phosphoribulokinase* (Fig. 5). Continuous transcriptional up-regulation extended to light-harvesting components including 14 genes encoding light-harvest complex (LHC) proteins F (LHCFs), one encoding LHCX1, 8 encoding LHCR, and 4 encoding fucoxanthin-chlorophyll *a/c* binding proteins (Fig. 5).

Six among 8 predicted hexose transporter encoding genes are transcriptionally down-regulated markedly with the addition of glucose relative to 0 h (Supplementary Fig. 10 and Supplementary Table 5). In co-culture, transcript levels of genes involved in upper glycolysis were not up-regulated when compared to those in algal monoculture except at 96 h. In addition, most genes involved in oxidative pentose phosphate pathway (OPPP) showed a trend of transcriptional down-regulation or no significant transcriptional changes (Fig. 5).

### Metabolite interaction between *P. tricornutum* and *L. vestfoldensis*

An untargeted metabolomic approach was used to analyze the supernatant metabolites of *L. vestfoldensis* monocultures (designated as "bacterial monoculture") and *P. tricornutum-L. vestfoldensis* cocultures (designated as "co-culture"). A total of 7337 characteristic peaks were detected in positive and negative ion patterns (Supplementary Data 5). PCA and partial least squares-discriminant analysis (PLS-DA) (Supplementary Fig. 11a, b) of all feature peaks and their contents indicated significant differences between bacterial monoculture and co-culture. Database comparison of MS1 resulted in a total of 1061 annotated compounds, while that of MS2 resulted in a total of 84 annotated compounds (Supplementary Data 6). Among the 84 metabolites, four metabolites, including isonicotinic acid, EDTA, loliolide, and P-anisic acid, increased significantly in co-culture; while

five metabolites, including 2-methylglutaric acid, adenosine, 4-hydroxy-6-methyl-2-pyrone, indole-3-acetic acid, and indole-3-carbinol decreased significantly (Fig. 6).

Cluster analysis of the 84 metabolites indicated that five main groups including 48 metabolites were closely clustered together (Blocks 1–5), in which Blocks 1 and 2 had a significant negative correlation with the metabolites in Blocks 4 and 5. Notably, isonicotinic acid, EDTA, loliolide, and P-anisic acid among the significantly increased metabolites belonged to Block 1, whereas the considerably decreased metabolites all belonged to Block 5 (Supplementary Fig. 11c and Supplementary Data 6).

## Discussion

Diatoms, responsible for approximately 20% of primary productivity on Earth, are one of the largest groups of marine phytoplankton[1,2]. However, inorganic carbon acquisition in seawater poses challenges to diatoms owing to the slow $CO_2$ diffusion in water, electrochemical gradients opposing the passive transmembrane transport of $HCO_3^-$, and the competition within phytoplankton communities[5]. Therefore, diatoms evolved sophisticated CCMs. In addition, a potential role of bacteria in providing organic carbon for the diatom growth is predicted in open oceans[10]. *P. tricornutum* is an obligate photoautotrophic diatom, and its growth strictly depends on the photosynthetically derived energy. All 12 *P. tricornutum* strains are unable to grow under darkness in our study, and glucose by itself does not support heterotrophic growth of this diatom, consistent with the previous findings[19]. Although some reports claimed that wild-type *P. tricornutum* could use exogenous glucose, illumination was a must for such utilization[20], which indicated that light supplied the energy and reducing equivalents. In our study, growth of the seemingly 'glucose-utilizing' strain PtCr needs light, and we identify it is the bacterium *L. vestfoldensis* that facilitates the growth of *P. tricornutum* when glucose is used as the sole carbon source. Adding *L. vestfoldensis* to the medium with glucose as the sole carbon source also allows Pt1-10 and UTEX640 to grow. Apparently, glucose cannot be directly utilized by all the 12 *P. tricornutum* strains. There are 8 predicted hexose transporter encoding genes in *P. tricornutum* genome. Seven (including a known vacuolar membrane-localized protein[21]) of them are not predicted to be a cell membrane-targeted proteins by the DeepLoc-2.1[22], and one (J9947) is

**Fig. 5 | CO₂ fixation pathways, photosynthesis, hexose transporters, and central carbon metabolism in *P. tricornutum* with the heatmap showing the gene expression changes in co-culture relative to algal monoculture.** OPPP oxidative pentose phosphate pathway, *CA* carbonic anhydrase, *LCIP63* carbonic anhydrase, *SLC4* solute carrier family 4, *PEPC* phosphoenolpyruvate carboxylase, *PEPCK* phosphoenolpyruvate carboxykinase, *MDH* malate dehydrogenase, *ME* malic enzyme, *PPDK* pyruvate phosphate dikinase, *HT* hexose transporter, *GLK* glucokinase, *GPI* glucose-6-phosphate isomerase, *PFK* 6-phosphofructokinase, *FBA* fructose-bisphosphate aldolase, *LHC* light harvest complex protein, *FCP* fucoxanthin chlorophyll a/c protein, *PsaO* photosystem I subunit O protein, *PsbM* photosystem II reaction center M protein, *PsbW* photosystem II W protein, *PsbO* photosystem II oxygen-evolving enhancer protein 1, *PsbP* photosystem II oxygen-evolving enhancer protein 2, *PsbQ* photosystem II oxygen-evolving enhancer protein 3, *Psb27* Photosystem II subunit 27, *PsbU* photosystem II extrinsic protein, *Psb31* Photosystem II subunit 31, *SBPase* sedoheptulose-1,7-bisphosphatase, *FBP* fructose-1,6-bisphosphatase, *RPI* ribose-5-phosphate isomerase, *PRK* phosphoribulokinase, *TAL* transaldolase, *TPI* triosephosphate isomerase, *PGK* phosphoglycerate kinase, *GAPDH* glyceraldehyde-3-phosphate dehydrogenase, *PGAM* phosphoglycerate mutase, *ENO* enolase, *PK* pyruvate kinase, *GPDH* glucose-6-phosphate dehydrogenase, *PGL* 6-phosphogluconolactonase, *PGDH* 6-phosphogluconate dehydrogenase, *TKL* transketolase, *RPE* ribulose-phosphate-3-epimerase. Asterisk indicates statistically significant differences of |log₂(fold change)| > 1 and *q* value < 0.05 (two-tailed unpaired *t*-test, 4 degrees of freedom, Benjamini-Hochberg correction, *n* = 3 biological replicates per group). 12, 24, 48, and 96 h, incubation time. Source data are provided as a Source data file.

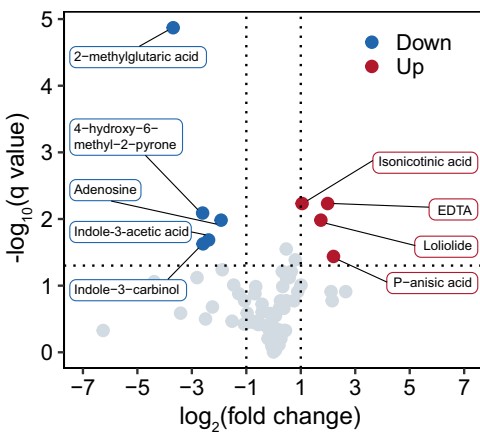

**Fig. 6 | Volcano plot showing significantly different metabolites (co-culture relative to bacterial monoculture, |log$_2$(fold change)| > 1, $q$ value < 0.05) in 84 metabolites from the MS2 data.** Co-culture, co-culture of *L. vestfoldensis* and *P. tricornutum*; Bacteria monoculture, monoculture of *L. vestfoldensis*. Down and Up represent metabolites with significant decrease and increase in content (two-tailed unpaired *t*-test, 4 degrees of freedom, Benjamini-Hochberg correction, $n$ = 3 biological replicates per group). Source data are provided as a Source data file.

predicted to be localized in cell membrane, lysosome, or vacuole, suggesting that extracellular glucose is unlikely to be transported into cells of wild-type *P. tricornutum*. Zheng et al.[20] show that transcription level changes of glucose transporters in *P. tricornutum* poorly correlate with the exposure to glucose, and suggest that these transporters shuttle glucose from the vacuole to the cytosol instead of transporting glucose into the cell.

Compared to that with atmospheric CO$_2$ as the sole carbon source, the growth of *P. tricornutum* with glucose as the sole carbon source is slower. The growth with atmospheric CO$_2$ and glucose as carbon sources is comparable to that with atmospheric CO$_2$ as the sole carbon source, indicating that an extra supplementation of glucose does not promote the algal growth. Furthermore, the algal growth is not affected by *J. anophelis* in the culture at least before day 6 when compared Pt1 with Pt1-sterile. However, interaction of *L. vestfoldensis* and PtCr is verified in our study. The absence of glucose in the media witnesses the growth of *L. vestfoldensis* dependent of the growth of the alga. When glucose is used as the sole carbon source, the bacterium grows better under light than under darkness, and it is indicated that algal growth is beneficial for the bacterium. When both organic glucose and inorganic CO$_2$ are available, bacterial growth is slightly inhibited by the strain PtCr. Our results suggest that under inorganic carbon limitation, algal cells obtain carbon sources generated by the bacterium through promoting the bacterial growth, thus accelerating its glucose utilization. On the contrary, when the inorganic carbon source is sufficient, algal cells inhibit the growth of the bacterium. The stimulatory effect of bacteria on diatom growth is not uncommon[23]. For example, the *Bacillus thuringiensis-P. tricornutum* co-culture showed a 2–3 fold increase in *P. tricornutum* cells at the stationary phase[24]. Co-cultures inoculated with a co-isolated Roseobacter supported the diatom *Pseudo-nitzschia subcurvata* growth in vitamin-limiting conditions[25], and bacteria in most natural symbionts can use organic carbon fixed by the algae[26]. In our study, however, mutualism between *L. vestfoldensis* and *P. tricornutum* occurs when the inorganic carbon source is limited for algae. To be specific, growth of the diatom depends on the support of *L. vestfoldensis* for the supply of necessary carbon source.

Although *J. anophelis* is also able to consume exogenous glucose, it does not support the growth of Pt1 when glucose is used as the sole carbon source. *Janibacter* sp., described as an algicidal bacterium, is reported to inhibit the growth of *Dunaliella* sp. in the inorganic carbon-replete medium[27]. Analysis of 180 macrogenomic datasets

from the global oceans[28] reveals that the frequency of diatoms co-occurring with *Janibacter* (0.04) is much lower than that with *Loktanella* (0.71) for all sampled sites. Co-occurrence analysis further supports the interactions between *Loktanella* and diatoms, including *Chaetoceros* and *Thalassiosira*, which has been validated in our co-culture experiments. In addition, Alphaproteobacteria, to which *L. vestfoldensis* belongs, are closely related to diatoms, as illustrated in many reports[18,25,29]. The representative genera, such as *Sulfitobacter* and *Roseobacter* have been frequently identified as symbiotic associates of diatoms[30–32]. *L. vestfoldensis* has also been isolated from another marine diatom *Skeletonema marinoi*[33], and shown to significantly enhance the algal growth under low iron concentrations, possibly by providing iron-chelating agents[34]. The long-term co-evolution of diatoms and *L. vestfoldensis* may bestow them a mechanism of interdependence for survival. We perceive *L. vestfoldensis* as a bacterial partner of strain PtCr rather than the experimental contamination.

Transcriptomic profiling reveals that HCO$_3^-$ transportation and dehydration are activated as shown by the up-regulation of PtCA1 and SLC4-2 when glucose is used as the sole carbon source. Transcript levels of genes encoding the membrane-localized SLC4-2, two pyrenoid-localized PtCAs, pyrenoid-penetrating thylakoid lumen-localized θCA, Calvin-Benson cycle enzymes, and light-harvesting components were markedly higher in co-culture conditions relative to algal monoculture, indicating algal cells are photoautotrophy-dependent. Furthermore, transcript levels of *glucokinase* (a hexokinase family member) are not changed, which suggests that glucose cannot be utilized by *P. tricornutum*. In contrast, the heterotrophic green alga *Chromochloris zofingiensis* shows strong induction of glycolysis upon glucose addition, with *hexokinase* transcripts increasing by ~40-fold within 1 h[35]. It is assumed that the carbon sources uptaken by *P. tricornutum* are CO$_2$ and low molecular weight carbon sources produced by *L. vestfoldensis* rather than glucose.

Non-targeted metabolomics reveals the difference in extracellular metabolites between the co-culture and bacterial monoculture. Metabolites with significantly lower concentrations in co-culture might be produced by *L. vestfoldensis* and utilized by *P. tricornutum*. Indole-3-acetic acid, a common metabolite produced and released by bacteria, has been proved to have the function of promoting algal cell growth[31,36], whereas indole-3-carbinol, 2-methylglutaric acid, 4-hydroxy-6-methyl-2-pyrone, and adenosine are little investigated in diatoms and hypothesized to potentially serve as carbon sources or signaling molecules. Substances released by some bacteria, such as biotin, thiamine, and vitamin B$_{12}$, can serve as nutrients for diatoms[25]. The acyl homoserine lactones, produced by bacteria and serving as signaling molecules, may bind to different targets in diatoms and initiate different responses that reflect whether these bacteria are synergistic or algicidal[11]. In contrast, the significantly increased metabolites are most likely produced by *P. tricornutum*; in which p-anisic acid and isonicotinic acid may have an antimicrobial activity[37,38], and loliolide (a carotenoid metabolite[39]) is reported as a biomarker for diatom productivity[40]. EDTA, a component of the medium, is significantly depleted in bacterial monoculture, suggesting that chelating agents secreted by bacteria are preferentially utilized by the diatom in co-culture. These significantly varied metabolites are classified into different blocks in the cluster analysis, and the correlation of metabolites within the blocks reflects that crucial metabolites interact with each other in co-culture and bacterial monoculture. The release of antimicrobial substances and organic matters by *P. tricornutum* and the release of growth-promoting factors and carbon sources by *L. vestfoldensis* suggest the interdependence and competition relationship between the bacterium and the diatom. This complex relationship also exists between *Croceibacter atlanticus* and *Pseudo-nitzschia multispectral*, and the bacterium colonises diatom cell surfaces and uses their exudates to proliferate until the diatom reaches the stationary growth phase[41].

The interaction between *P. tricornutum* and *L. vestfoldensis*, which uniquely emerges in PtCr among 12 ecologically distinct diatom strains, exhibits the high specificity. This interaction contrasts sharply with the antagonistic relationship observed between Pt1 and *J. anophelis* when algal growth actively suppresses bacterial proliferation. Here, we propose a dynamic carbon-dependent interaction model modulated by environmental carbon availability (Supplementary Fig. 12). When inorganic carbon is sufficient with a lack of organic carbon, *P. tricornutum* prioritizes autotrophic growth while provisioning organic substrates to sustain *L. vestfoldensis*. Conversely, inorganic carbon limitation with organic carbon sufficiency triggers bacterial processing of glucose into $CO_2$ and growth stimulants (e.g., indole-3-acetic acid), thus fueling diatom metabolism in return; while the organic matters yielded by *P. tricornutum* further promote the growth of *L. vestfoldensis*. Dual carbon availability induces competitive dynamics: *P. tricornutum* preferentially utilizes inorganic carbon while partially suppressing *L. vestfoldensis* via antimicrobial secretion (e.g., p-anisic acid). In open ocean surface water, the concentration of carbohydrates is approximately 10–25 μM C glucose equivalent with a glucose concentration of around 100 nM, making carbohydrates the most abundant identified component of dissolved organic matter in seawater[42,43]. Although inorganic carbon is relatively abundant in surface seawater at millimolar concentrations[5], the sharp decrease of inorganic carbon in the environment (e.g., during intense phytoplankton blooms or in regions of restricted deep water mass formation and suppressed vertical mixing) may associate obligate photoautotrophic diatoms tightly with bacteria and urge the algae to use carbon sources deriving from carbohydrates. Our findings interpret the bacterial mediation of diatom mixotrophic adaptation. The multifaceted interaction illustrates the adaptive microbial partnership that balances mutualism and competition in fluctuating marine carbon regimes. These findings advance our understanding of the marine carbon cycle and provide a basis for a general interaction model. However, the current model, established under simplified laboratory system, remains preliminary and requires further validation to determine its ecological relevance. Future work will focus on uncovering key molecular mechanisms and validating the model using environmentally relevant conditions and field data.

# Methods

## Algal strains and culture conditions

The twelve *P. tricornutum* strains and *C. muelleri* used in this study were obtained from six culture collections, namely Provasoli-Guillard National Center for Culture of Marine Phytoplankton (CCMP; Pt1 = CCMP632, Pt5 = CCMP630, Pt6 = CCMP631, Pt7 = CCMP1327, Pt9 = CCMP633, and *C. muelleri* = CCMP1316), Culture Collection of Algae and Protozoa (CCAP; Pt2 = CCAP 1052/1 A, Pt3 = CCAP 1052/1B, Pt4 = CCAP 1052/6), Canadian Center for the Culture of Microorganisms (CCCM; Pt8 = NEPCC 640), Microalgae Culture Collection of Qingdao University (MACC; Pt10 = MACC B228), Institute of Oceanography of Chinese Academy of Sciences (PtCr = CCMM 2004), University of Texas at Austin Collection (UTEX640) (Supplementary Fig. 13). *T. pseudonana* (CCMP1335) was kindly provided by Professor Thomas Mock at University of East Anglia (UK). The internal transcribed spacer (ITS) region of strains PtCr and UTEX640 was amplified using PCR with primer pair TW13 (5′-GGTCCGTGTTTCAAGACG-3′) and ITS3 (5′-GCATCGATGAAGAACGCAGC-3′) according to De Martino et al.[44], and a phylogenetic tree was generated using maximum likelihood algorithm based on the ITS sequences of the 12 *P. tricornutum* strains.

All strains were cultured in artificial seawater supplemented with f/2 nutrient[45] (without silicate except for *C. muelleri* and *T. pseudonana*) at 22 °C with continuous LED illumination (100 μmol photons m$^{-2}$ s$^{-1}$) and with shaking at 120 rpm. Prior to batch cultivation, the strains were treated with 20 mg/L ampicillin and 20 mg/L kanamycin to remove bacteria (standard antibiotic treatment), and 20 mg/L kanamycin was added to inhibit bacterial growth in all cultivation experiments. To eliminate the bacteria thoroughly, Pt1 was cultivated in f/2 enriched medium supplemented with 20 mg/L ciprofloxacin and 10 mg/L streptomycin for 5 days, and then the diluted culture was coated onto f/2 agar plates with the two antibiotics for two weeks (intensive antibiotic treatment). A single algal colony was picked onto the LB agar plate to check the bacterial contamination, and obtain sterile Pt1 strain (Pt1-sterile).

## Batch culture

Twelve *P. tricornutum* strains were cultured in carbon-free (without NaHCO$_3$) artificial seawater supplemented with f/2 nutrient (with or without silicate) at an initial density of $2 \times 10^5$ cells/mL. Incubation was carried out in vent cap (AC, MC) or plug-seal cap (CL, GC) culture flasks with (MC, GC) or without (AC, CL) the supplement of 3 g/L glucose on an orbital shaker. Thus, AC stands for atmospheric $CO_2$ as the sole carbon source ($CO_2$ supply through passive diffusion from the air; preliminary experiments showed addition of 2 mM NaHCO$_3$ in the medium did not promote the growth of *P. tricornutum*, indicating inorganic carbon was enough under AC condition) (Supplementary Fig. 14), MC for atmospheric $CO_2$ and glucose as carbon sources, CL for carbon-limited conditions, and GC for glucose as the sole carbon source. The pH and total inorganic carbon were measured using a pH meter (Delta320, China) and a TOC analyzer (Vario TOC cube, Germany), respectively.

To further confirm growth phenotypes, Pt1, Pt1-sterile (as the control), and the 'glucose-utilizing' strain PtCr were cultured under AC, MC, and GC conditions with continuous light and cultured under darkness with atmospheric $CO_2$ and glucose as carbon sources (MC-Dark) or with glucose as the sole carbon source (GC-Dark). Cultures (30 mL) were maintained in an incubator (ZQZY-75AGN, China). Every two days, cell density was quantified by microscopic counting, and glucose concentration was measured spectrophotometrically using a commercial assay kit (Beyotime, China) following the manufacturer's protocol. All experiments were performed in triplicate. A schematic diagram of the experimental design is shown in Supplementary Fig. 15.

## Bacterial isolation and identification

Cultures of PtCr or Pt1 grown for 6 days under MC-Dark conditions were filtered using glass microfiber filters (Whatman GF/C 1.2 μm, USA). The filtrate was transferred to 48-well plates and incubated in f/2-enriched artificial seawater with glucose as the sole carbon source for 2 weeks. Bacterial isolates were identified through *16S rRNA* gene sequencing using universal primers 16s-27F (5′-AGAGTTT-GATCCTGGCTCA-3′) and 16s-1492R (5′-GGCTACCTTGTTACGACTT-3′) by PCR with an annealing temperature of 58 °C[46]. Amplified products were sequenced and analyzed using the Basic Local Alignment Search Tool (BLAST) against the National Center for Biotechnology Information (NCBI) nucleotide database for taxonomic classification. The sequences of 16S rRNA Sanger sequencing of Pt1 isolate *J. anophelis* and PtCr isolate *L. vestfoldensis* are archived in GenBank under accession numbers PX410800 and PX410801, respectively.

To validate the function of *L. vestfoldensis*, the epiphytic bacterium isolated from PtCr cultures, 1 mL of the bacterial culture (OD$_{600}$ ≈ 0.4) was added to the cultures of the 12 *P. tricornutum* strains with initial cell density of $2 \times 10^5$ cells/mL, and 1 mL of sterile artificial seawater was used as the control. Cultivation was performed under GC conditions as mentioned above.

## Detection of bacterial abundance in PtCr culture

Strain PtCr was cultured under AC, CL, MC, and GC conditions, together with AC-Dark (atmospheric $CO_2$ as the sole carbon source under darkness), CL-Dark (carbon-limited conditions under darkness), MC-Dark, and GC-Dark conditions. Samples were collected every two days, and total DNA was extracted. The relative abundance of the epiphytic

bacterium was quantified through *16S rRNA* gene amplification using quantitative PCR (qPCR) with primer pair Cr-16s-qpcr-F (5′-GCGGATTGGAAAGTATGGG-3′) and Cr-16s-qpcr-R (5′-TCGCACCT-CAGCGTCAGTA-3′). The qPCR reactions were performed using PerfectStart® Green qPCR SuperMix (TransGen Biotech, China) following the manufacturer's protocol.

## 16S rRNA amplicon sequencing of epiphytic bacteria

All the diatoms obtained from the culture collections were cultured for 8 days, and then algal samples were collected by centrifugation for total DNA extraction. The total DNA of each diatom was used to amplify the V3-V4 region of *16S rRNA* gene using universal primer 338 F (5′-ACTCCTACGGGAGGCAGCA-3′) and 806 R (5′-GGAC-TACHVGGGTWTCTAAT-3′). The PCR amplicons were purified with Agencourt AMPure XP Beads (Beckman Coulter, USA) and quantified using the Qubit dsDNA HS Assay Kit and Qubit 4.0 Fluorometer (Thermo Fisher Scientific, USA). Qualified amplicons were used to construct libraries and sequenced using Illumina novaseq 6000 (Illumina, USA) with 150 bp paired-end reads. The sequencing data have been deposited in the Sequence Read Archive database under accession number PRJNA1330494. Raw data were primarily filtered to obtain clean reads by Trimmomatic (version 0.33) and Cutadapt (version 1.9.1)[47,48]. Clean reads were assembled and followed by chimera removal using DADA2[49]. Sequences with similarity ≥ 97% were clustered into the same operational taxonomic unit (OTU) by VSEARCH (version 2.26.1)[50], and the OTUs with abundance < 5 were filtered. We used BLAST to remove OTUs with similar sequences to the mitochondria (NCBI nucleic acid numbers NC_016739.1, LC537470.1, and NC_007405.1) and chloroplast genomes (NC_008588.1, NC_053621.1, and NC_008589.1) of *P. tricornutum*, *C. muelleri*, and *T. pseudonana*, with a filtering condition of similarity > 90%, e-value < 10⁻⁵. Taxonomy annotation of the OTUs was performed based on the Naive Bayes classifier in QIIME2[51] using the SILVA database (release 138)[52] with a confidence threshold of 70%. R (version 4.2.1) with the ggplot2 package (version 3.5.1)[53] was used for data analysis and visualization.

## Distribution and co-occurrence of epiphytic bacteria and diatoms in the global ocean

Based on species diversity data from 180 macrogenomes in the *Tara* Oceans database[28], the abundance of diatoms, *Janibacter*, and *Loktanella*, and the bacterial-algal co-occurrence frequency were analyzed in global oceans. Geographic visualizations used the ggplot2 package and public-domain map data from the maps package (version 3.4.0)[54]. The metagenomic data analyzed in this study originated from the *Tara* Oceans metagenomic datasets (https://www.ebi.ac.uk/biostudies/files/S-BSST297/OM-RGC_v2_taxonomic_profiles.tar.gz).

Co-occurrence network inference was performed using FlashWeave (version 0.18.1)[55] with Julia (version 1.4.2), for *Loktanella* and *Janibacter* linked to diatoms. The data of the genus level is sourced from the *Tara* Oceans metagenomic datasets. FlashWeave parameters used were sensitive = true, heterogeneous = false, alpha = 1.0, n_obs_min = 0.

## RNA extraction, transcriptome sequencing, and data analysis

Strain Pt1 (initial density of $2 \times 10^6$ cells/mL) was cultured with 3 g/L glucose as the sole carbon source (GC), and 1 mL of the PtCr epiphytic bacterium culture ($OD_{600} \approx 0.4$) was added to the Pt1 culture (designated as "co-culture"), with 1 mL of sterile artificial seawater (without $NaHCO_3$) as the control (designated as "algal monoculture"). Cultures were collected at 0, 12, 24, 48, and 96 h, washed twice with phosphate-buffered saline, flash-frozen in liquid nitrogen, and stored at −80 °C. Total RNA was extracted using TRIzol reagent (TransGen Biotech, China), and the purity and integrity of RNA samples were assessed using a NanoPhotometer® spectrophotometer (Implen, Germany) and Agilent 2100 RNA Nano 6000 Assay Kit (Agilent Technologies, USA).

Qualified samples were used to construct sequencing libraries using the NEBNext® Ultra™ RNA Library Prep Kit (Illumina, USA) and sequenced using Illumina HiSeq 2000 (Illumina, USA) with 150 bp paired-end reads in three replicates per sample. The transcriptomic data have been deposited in the Sequence Read Archive database under accession number PRJNA1276728.

Raw reads were filtered using Fastp (version 0.12.4) to remove adapters and low-quality reads[56]. Clean reads were mapped to the reference genome of *P. tricornutum* using Hisat2 (version 2.1.0)[57,58]. FeatureCount (version 2.0.1) was used to obtain read counts for each gene and manually calculate reads per kilobase of transcript per million mapped reads (RPKM)[59]. Gene expression was normalized using the R package DESeq2 (version 1.28.1)[60]. PCA and cluster analysis were performed on normalized data. DEGs were identified using stringent thresholds ($|\log_2(\text{fold change})| > 1$, $q$ value < 0.05) comparing co-culture and algal monoculture conditions. The number of up- and down-regulated DEGs was counted, and pathway enrichment analysis was performed for DEGs based on the Kyoto Encyclopedia of Genes and Genomes (KEGG) using the R package clusterProfiler (version 3.16.1)[61]. Transcriptional profiles of key metabolic pathways were analyzed, including CCMs, Calvin cycle, photosynthesis, glycolysis/gluconeogenesis, and OPPP, with visualization performed using ggplot2 package in R.

## Metabolomic sample preparation and data analysis

One mL ($OD_{600} \approx 0.4$) of PtCr epiphytic bacterium was inoculated into 50 mL plug-seal cap culture flasks containing 30 mL of f/2-enriched artificial seawater (without $NaHCO_3$) with 3 g/L glucose as the sole carbon source, and then Pt1 ($6 \times 10^6$ cells) (co-culture) or bicarbonate-free artificial seawater (designated as "bacterial monoculture") of an equal volume was added to the bacterial cultures. Subsequently, the cultures were maintained at 22 °C with shaking (120 rpm) under continuous illumination of 100 µmol photons $m^{-2} s^{-1}$ for 6 days. All cell cultures were centrifuged at $3000 \times g$ for 10 min, and 30 mL supernatant was collected and stored at −80 °C.

For LC-MS analysis, 1 mL of samples was centrifuged at 4 °C, $13,000 \times g$ for 5 min, then the supernatant was collected and filtered through a 0.22 µm aqueous membrane. 80 µL of filtrate and 20 µL of 300 µg/mL internal standard were added for sample analysis. A total of 11 samples were analyzed, including 2 experimental samples (each with 3 biological replicates), 1 quality control sample (with 4 replicates), and 1 blank sample of methanol. Quality control samples came from a mixture of the experimental samples. The quality control sample of 200 µL was divided into four, then three (i.e., pooled samples) were used to monitor the precision of the instrument, and one was used to monitor the stability of the instrument. Metabolite profiling was performed using an Ultimate 3000 UHPLC system (Thermo Fisher Scientific, USA) coupled to a Q Exactive Orbitrap mass spectrometry (Thermo Fisher Scientific, USA) equipped with a Waters ACQUITY UPLC HSS T3 column (Waters, USA; 100 mm × 2.1 mm × 1.8 µm). Both positive and negative ionization modes were employed for comprehensive metabolite detection. Mobile phases consisted of: (1) positive mode−0.1% formic acid in water (A) and 0.1% formic acid in methanol (B); (2) negative mode−10 mM ammonium formate in water (A) and 10 mM ammonium formate in 95% methanol (B). The gradient program was: 0−1 min, 10% B; 1−13 min, linear increase to 98% B; 13−18 min, 98% B; 18−18.5 min, linear decrease to 10% B; 18.5−20 min, 10% B. The injection volume for each sample was 2 µL, and methanol was used as a blank sample to deduct baseline features from the mobile phase. The mass spectrometer was operated in a data-dependent acquisition (DDA) mode with positive and negative ions scanned in separate runs. The acquisition cycle consisted of a full MS scan at a resolution of 70,000, followed by data-dependent MS2 (dd-MS2) scans of the most intense precursors at a resolution of 17,500. Electrospray ionization (ESI) source parameters were set as follows:

spray voltages, 3.8 kV for positive mode and 3.2 kV for negative mode; capillary temperature, 300 °C; sheath gas flow rate, 0.3 mL/min; and nebulizer temperature, 350 °C. Samples were maintained at 4 °C in the autosampler during analysis.

Raw LC-MS data were converted to Analysis Base File (ABF) format using the ABF Converter tool. MS-DIAL software (version 4.70)[62] was used to perform blank deduction (peak intensity [Sample/Blank] ≥ 3), feature peak screening (at least 2 feature peaks appearing in repeated samples), and normalization (LOWESS method) on the data of positive and negative ion modes separately. A comprehensive feature matrix was generated, containing metabolite identifiers, RT, mass-to-charge ratio (m/z), ion patterns, and peak intensities. Metabolite annotation was achieved by matching experimental RT and m/z values against the MassBank database with the following parameters: RT tolerance = ±0.1 min, MS1 mass tolerance = 0.005 Da, MS2 mass tolerance = 0.0025 Da, and MS2 spectral match score > 0.7. Missing values were imputed using 20% of the minimum value within each experimental group[63]. The metabolomics data with Study ID ST004351 are available at the NIH Common Fund's National Metabolomics Data Repository (NMDR) website, the Metabolomics Workbench (https://www.metabolomicsworkbench.org)[64].

For data analysis, peak intensities were subjected to Pareto scaling and $log_{10}$ transformation to minimize magnitude-related artifacts[65]. Multivariate statistical analyses, including PCA, PLS-DA, and hierarchical clustering, were performed using R and R package mixOmics (version 6.24.0)[66] to evaluate intergroup variability, metabolite correlations, and differential features. Differentially abundant metabolites under co-culture and bacterial monoculture conditions were identified using Student's t-test, with stringent thresholds of $|log_2(fold change)| > 1$ and a q-value < 0.05 (Benjamini−Hochberg adjusted p value).

## Statistics and reproducibility

Sample sizes are based on similar studies published previously and are sufficient to show statistical differences between the experimental group and the control group. No data are excluded from the analysis. All the data are based on at least 3 independent biology replicates to ensure the reliability of the results, and the number of replicates is indicated in the corresponding methods section. Samples are selected randomly. Noblinding is done as none of the experiments described in this study involves group allocation during data collection or analyses.

## Reporting summary

Further information on research design is available in the Nature Portfolio Reporting Summary linked to this article.

## Data availability

The raw data of RNA-Seq and 16S rRNA amplicon sequencing have been deposited in the Sequence Read Archive database under accession numbers PRJNA1276728 and PRJNA1330494. The Sanger sequencing data of 16S rRNA have been deposited in the Genbank database under accession numbers PX410800 (*Janibacter anophelis*) [https://www.ncbi.nlm.nih.gov/nuccore/PX410800] and PX410801 (*Loktanella vestfoldensis*) [https://www.ncbi.nlm.nih.gov/nuccore/PX410801]. The metabolomic data have been deposited on Metabolomics Workbench under Study ID ST004351 [https://doi.org/10.21228/M8WR9Z]. Source data are provided with this paper.

## Code availability

The code used to produce the results is available at Figshare (https://doi.org/10.6084/m9.figshare.30178000).

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

## Acknowledgements

This work was funded by the National Key Research and Development Program of China with No. 2023YFA0914400 (2023YFA0914402 to H.H.) and National Natural Science Foundation of China with No. 42376131 (to H.H.). We acknowledge the Metabolomics Workbench (https://www.

metabolomicsworkbench.org, supported by NIH grants U2C-DK119886 and OT2-OD030544) for hosting the data and metadata for this study (Study ID: ST004351).

## Author contributions

H.H. designed the research. C.L., W.Y., and Y.P. performed experiments and data analysis. C.L. and H.H. drafted the manuscript. H.H. revised the manuscript. H.H. acquired the financial support. All authors read and approved the final manuscript.

## Competing interests

The authors declare no competing interests.
