## [Transparent Peer Review file · Nature Communications]

Interactions with bacteria shape diatom adaptation to carbon concentration changes

Corresponding Author: Dr Hanhua Hu

Version 0:

Reviewer comments:

Reviewer #1

(Remarks to the Author)

Overview:

In this study, Li et al. performed a series of laboratory culture experiments with diatom strains and bacteria and explored physiological and functional responses among these microbes to various conditions, which included addition of an exogenous carbon source, inorganic CO₂, and dark vs. light treatments. Importantly, they found evidence of mutualistic relationships between diatoms and bacteria in culture, where the bacteria depended on the diatom when CO₂ was the only carbon source and diatom growth relied on the presence of the bacteria when glucose was the sole carbon source. They used multiple 'omics methods (transcriptomics and metabolomics) that revealed important insights into the functional responses of diatom-bacterial relationships and supported some of their other observations. Lastly, they put their culture work into a broader context by including some information from Tara Oceans samples that confirmed the overlapping niche of these organisms at scale in the field.

Overall, I enjoyed the manuscript and feel that is a strong contribution and should be considered for this journal. However, I do have some suggestions (many of them minor) that I feel may improve the paper. Below are specific comments and suggestions with line references.

Has the code associated with this work been archived somewhere, specifically the code used to make the figures for your 'omics results? I may have missed this, but it should be either on GitHub or a similar repository.

Specific comments:

Line 20: What do you mean by redistribution?

Lines 27-30: I feel like these two sentences about their potential for biofuel and other resources comes a bit out of left field. Maybe add a bridge sentence or some more information first about their dynamics in the global oceans and maybe their environmental drivers?

Line 47: As in they consume bacteria? I think when you mention mixotrophic you are referring to consumption of carbon and not other cells. You get to that in the next section, but I would specify potential carbon consumption here to avoid confusion.

Line 70: Is reanalysis a better way to phrase this? Basically, you mined the dataset and looked for these taxa, correct? But did you do any new analysis with the dataset?

Lines 70-73: This last statement seems a little strong. Maybe scale it back a little and mention that your work reveals novel insights into (insert those novel insights) that provides a basis for explaining the adaptation of marine diatoms to inorganic carbon-limited environments. Highlighting the mutualistic interactions that you verified in the lab and under different conditions.

Line 76: Rephrase? The start of the sentence reads a bit off.

Lines 94-96: So, they could consume glucose, but this did not get allocated to observable population growth? Am I following that correctly?

Lines 96-99: But weren't these cultures axenic? Or that was as separate experiment that comes in later?

Lines 125-127: Did you do a co-occurrence analysis? Or this is just based on the presence of these taxa and how that overlaps in the dataset? I would be careful with co-occurrence or just explain this in more detail in the methods.

Line 193: Did you analyze negative and positive mode features separately or merge them?

Line 290: Consider changing metabolome to metabolomics.

Line 342: It may be nice to have a few sentences to really drive home the broader implications of this work and some of the next steps in the field and lab. For instance, getting to a point of modeling these interactions in the ocean with changing conditions.

Line 412: It might be nice to add a bit more detail here. Was this based on occurrence of these taxa of interest in the Tara Oceans data table? Which specific resource did you use? Link to that resource? Was this from the metagenome or metabarcoding data? Or does macrogenome encompass all these samples?

Line 432: What about the other 16S sequence data? Has that been archived?

Line 443: Here and elsewhere in the methods, I would consider adding a citation for R packages that you use, if they are available.

Line 455: How much of the supernatant did you collect? Was it roughly the same across samples and replicates?

Line 456: How much volume from each sample for this pooled sample? Did you run the pooled sample once or multiple times on the mass spec?

Line 469: How much sample volume did you inject? Did you have instrument blanks to account for baseline features in the mobile phase? Just a few more details here would be nice. Also, is this metabolomics data archived somewhere (e.g., MetaboLights)?

Line 472: Citation for MS-DIAL?

Line 479: Did you normalize the feature intensities across samples? For instance, normalizing based on volume or weight. Did you do any other filtering steps, like removing a feature if not found in 2/3 replicates? Or subtracting the intensity from the blanks?

Line 486: These are more so curiosities or ideas for next steps that may be nice to consider. It would be helpful to have a host-only treatment as well, so that you could really see what features are specific to the co-culture treatment. Also, I would be interested to run a targeted approach to look at some of these compounds in more detail and/or provide more context to the transcriptome results. These are just cool next steps, that you could mention in the discussion section or expand upon.

Figure 3: Can you give more details in the legend about what the circles and their sizes represent. I know it is OTU abundance, but this should be specified in the legend. It would also be nice to explain the insert bar chart. Can you make that bar chart larger? It is difficult to read the axes labels.

Figure 5: More details in the legend would be nice. Is this showing just the 84 compounds from the MS2 data? This is not all the features, right? I would specify that.

Reviewer #2

(Remarks to the Author)

Diatoms are important primary producers, and the interaction between diatoms and their associated bacteria are vital for the marine ecosystem's productivity, nutrient cycling, and overall biogeochemical processes. The authors isolated *Loktanella vestfoldensis* from a culture of the model diatom *Phaeodactylum tricornutum* and found that its interaction with *P. tricornutum* was dependent on the carbon source. *L. vestfoldensis* showed dependence on *P. tricornutum* when CO₂ was the sole carbon source, and growth of the diatom depended on the support of *L. vestfoldensis* for the supply of necessary carbon source when glucose served as the sole carbon source. Genes involved in inorganic carbon concentration, Calvin cycle, and photosynthesis were found to be upregulated in the co-culture of *L. vestfoldensis* and *P. tricornutum*.

Co-culture of *L. vestfoldensis* and *P. tricornutum* transcriptionally upregulated genes involved in the inorganic carbon concentration, Calvin cycle, and photosynthesis, while upper glycolysis remained unaffected. These findings highlight the importance of bacterial-algal interactions in diatom carbon utilization and contribute to marine carbon cycling and redistribution.

The experiments were thoughtfully designed, and the manuscript is well-written. The findings of this study hold significant

implications for the field of phytoplankton ecology. However, certain aspects could be refined to better articulate the scientific questions and improve overall clarity for readers.

1. Line 31: "Depletion of inorganic carbon in the surface seawater, compounded by intense phytoplankton photosynthesis, poses constraints to the growth of diatoms."

In surface seawater, dissolved inorganic carbon (DIC) primarily exists as bicarbonate (HCO_3^-), aqueous CO_2 , and carbonate (CO_3^{2-}). Among these, bicarbonate dominates the DIC pool due to its high concentration, while dissolved CO_2 remains comparatively low because of its limited solubility under surface ocean conditions. Please rephrase this sentence.

2. Line 372 : Incubation was carried out in vent cap (AC, MC) or plug-seal cap (CL,GC) culture flasks with (MC, GC) or without (AC, CL) the supplement of 3 g/L glucose. Thus, AC stands for atmospheric CO_2 as the sole carbon source, MC for atmospheric CO_2 and glucose as carbon sources, CL for carbon-limited conditions, and GC for glucose as the sole carbon source.

While the abbreviation 'MC' was properly defined in the Methods section, its first occurrence in the Results (Line 84) lacks a parenthetical reminder of its meaning. For optimal readability, we recommend either: reintroducing the abbreviation parenthetically when first used in later sections (e.g., MC: mixed carbon)

3. To clarify the experimental treatments:

Pt1: Received standard antibiotic treatment to partially remove associated bacteria.

Pt1 sterile: Underwent intensive antibiotic treatment to achieve complete bacterial removal.

Pt Cr: Served as an untreated control, maintaining its native bacterial community.

While these protocols were described in the Methods, we suggest explicitly restating this critical distinction in the Results or Discussion to enhance reader comprehension of the experimental design

4. I wonder *L. vestfoldensis* is the only bacteria species you isolated from the *P. tricornutum* culture?

5. Line 397: "To validate the function of *Loktanella vestfoldensis*, the epiphytic bacterium isolated from PtCr cultures, 1 mL of the bacterial culture ($\text{OD}_{600} \approx 0.4$) was added to the cultures of the 12 *P. tricornutum* strains, and 1 mL of sterile artificial seawater used as the control. Cultivation was performed under GC condition as mentioned above."

How about the initial cell concentration of *P. tricornutum* in the co-culture?

Reviewer #3

(Remarks to the Author)

The manuscript addresses the question of species interactions between diatoms and bacteria. The authors picked the model *P. tricornutum* and two co-occurring bacteria from laboratory cultures to investigate as to how inorganic vs organic carbon supply impacts their interactions under light and darkness and in the presence and absence of the bacteria.

Although I think this research is timely, I had some issues with this manuscript. First, I struggled to understand the main results based on reading the abstract. Hence, I think the authors do need to be doing a better job in summarising the main results in a concise and accessible way. Even though I continued reading, I still think the authors should be adding a schematic diagram that illustrates the experimental setup and how the different strains of the diatom have been used. Although the results are interesting and worth following up on, I have methodological and conceptual issues with the presented work.

Conceptual issues:

A) I think picking *P. tricornutum* as the diatom model was a bad choice considering what the overarching question is. My reasons are as follows: 1) The authors have not provided any evidence beyond culture conditions that *P. tricornutum* is associated with either of the two bacterial species in nature. The lab is a very different environment and this therefore has to be appreciated. To address this concern, I suggest to do co-occurrence analyses to see which are the main diatom species associated with both bacterial species in the oceans opposed to the lab. This can be done by using the TARA Oceans data, and to be honest, this should have been done before setting up any experiments. Some of those data are already available to the authors because they should know from generating figure 3 which are the associated diatom species. 2) I am sure that *P. tricornutum* does not only associate with two different bacterial species in laboratory cultures. To address this issue, this work needs to explore 16S amplicon sequencing for every diatom strain used in this study to comprehensively assess all interacting bacteria in the laboratory cultures.

B) How meaningful are experiments with marine photosynthetic organisms in an environment that does not contain bicarbonate? The latter is the main source of DIC (Dissolved inorganic carbon) in the oceans for many photosynthetic organisms including diatoms (e.g., through using an effective carbon concentrating mechanism). Thus, experiments that do not at least control for the presence of bicarbonate, miss out on ecological relevance and therefore wider significance and applicability of the results.

Methodological issues:

A) Provide evidence of axenic diatom cultures. As no evidence has been provided, I doubt that those cultures have been grown under axenic conditions although the results are different between the axenic vs xenic experiments. But this needs to be clarified, nevertheless.

B) I have not seen any robust evidence that the associated bacteria are symbionts (line 317).

C) No information has been given on the CO₂ concentrations. I have not even seen how CO₂ has been added. Were the cultures bubbled with air? Please provide details and CO₂ concentrations.

D) With referring to B) under conceptual issues, what was the pH of the cultures? Was there a buffer used? The pH has a significant impact on the growth of diatoms and overall metabolism. Thus, pH data need to be provided.

E) Why was no silicate provided? This adds to concerns of wider applicability to the natural system as most diatoms require silicate to grow. *P. tricornutum* appears to be the odd ball out.

Version 1:

Reviewer comments:

Reviewer #1

(Remarks to the Author)

The authors have implemented my suggested edits and I have no further concerns with the manuscript at this time. Thanks for all of your hard work on improving the paper!

Reviewer #2

(Remarks to the Author)

Thank you for revising the manuscript in response to the reviewers' comments and suggestions. From my perspective, you have fully addressed the questions I raised in the first round of review.

Reviewer #3

(Remarks to the Author)

The revised version of this manuscript has addressed all of my comments. The additional experiments and data have made the results more robust and relevant for a broader community of scientists. Weaknesses have been addressed appropriately. Congrats to this thorough revision! I have only some minor suggestions to improve the manuscript before it can be accepted. I do not need to see the next version anymore.

- Figures 1-4 miss to mention the number of replicates (N=?) in the figure legends. Also, actual exact values for "p" should be provided and the test that has been used to calculate them.

- Figure 2 has too much redundant information. I think the level of organisation that matters the most considering the rest of the results is the genus level. Hence, move panels a, b, and c to the supplement.

- I don't think figure 7 is necessary. The information presented in this figure is simplistic and has already been well described and discussed in the text of the manuscript. Either delete it or move it to the supplement. Another reason: similar cartoons have been published before. What maybe would make sense is to extend existing concepts of diatom-bacteria interactions with the new data presented in this manuscript.

Reviewer #1 (Remarks to the Author)

Overview:

In this study, Li et al. performed a series of laboratory culture experiments with diatom strains and bacteria and explored physiological and functional responses among these microbes to various conditions, which included addition of an exogenous carbon source, inorganic CO₂, and dark vs. light treatments. Importantly, they found evidence of mutualistic relationships between diatoms and bacteria in culture, where the bacteria depended on the diatom when CO₂ was the only carbon source and diatom growth relied on the presence of the bacteria when glucose was the sole carbon source. They used multiple omics methods (transcriptomics and metabolomics) that revealed important insights into the functional responses of diatom-bacterial relationships and supported some of their other observations. Lastly, they put their culture work into a broader context by including some information from Tara Oceans samples that confirmed the overlapping niche of these organisms at scale in the field.

Overall, I enjoyed the manuscript and feel that is a strong contribution and should be considered for this journal. However, I do have some suggestions (many of them minor) that I feel may improve the paper. Below are specific comments and suggestions with line references.

Has the code associated with this work been archived somewhere, specifically the code used to make the figures for your 'omics results? I may have missed this, but it should be either on GitHub or a similar repository.

Reply: We thank the reviewer for this suggestion. The code used for omics analysis and figure generation has been deposited in a Figshare repository (<https://figshare.com/s/d84ef77b66aa6082d1a8>) and is now cited in the Code Availability section.

Specific comments:

Line 20: What do you mean by redistribution?

Reply: We have deleted the word “redistribution” in the revised abstract. Usually, bacteria obtain organic carbon from diatoms, and we used “redistribution” to emphasize that diatoms obtain the carbon source for growth from bacteria.

Lines 27-30: I feel like these two sentences about their potential for biofuel and other resources comes a bit out of left field. Maybe add a bridge sentence or some more information first about their dynamics in the global oceans and maybe their environmental drivers?

Reply: We have removed these two sentences as suggested and added the following sentence: “Due to siliceous cell walls, diatoms are prone to sedimentation and form a net carbon flux towards the lower layer of seawater, and are considered the main participants in the carbon pump of marine organisms.” (Lines 30-32).

Line 47: As in they consume bacteria? I think when you mention mixotrophic you are referring to consumption of carbon and not other cells. You get to that in the next section, but I would specify potential carbon consumption here to avoid confusion.

Reply: We have clarified that in the revised manuscript (Lines 50).

Line 70: Is reanalysis a better way to phrase this? Basically, you mined the dataset and looked for these taxa, correct? But did you do any new analysis with the dataset?

Reply: We have changed it to “reanalysis” as suggested. We used *Tara* Oceans macrogenomic data to analyze the overlap of diatoms and two bacteria (*Loktanella* and *Janibacter*) at different sites. In the revised manuscript, we have conducted co-occurrence analysis on the three groups (a new figure, Fig. 4c).

Lines 70-73: This last statement seems a little strong. Maybe scale it back a little and mention that your work reveals novel insights into (insert those novel insights) that provides a basis for explaining the adaptation of marine diatoms to inorganic carbon-limited environments. Highlighting the mutualistic interactions that you verified in the lab and under different conditions.

Reply: We have toned down the conclusion as suggested, by stating: “Our study reveals novel insights into the interaction between bacteria and diatoms under different availability of inorganic and organic carbon, and provides a basis for explaining the adaptation of marine diatoms to inorganic carbon-limited environments.” (Lines 75-77).

Line 76: Rephrase? The start of the sentence reads a bit off.

Reply: We have rephrased the sentence as suggested (Lines 82-83).

Lines 94-96: So, they could consume glucose, but this did not get allocated to observable population growth? Am I following that correctly?

Reply: Exactly. As shown in Supplementary Fig. 3, though glucose consumption was detected in cultures of both Pt1 and PtCr under darkness, the two Pt strains did not grow. Neither the consumption of glucose nor growth was observed in Pt1-sterile culture. This suggests that glucose consumption is unlikely to be due to heterotrophic growth of *P. tricornutum*, but rather results from the epiphytic bacteria of Pt.

Lines 96-99: But weren't these cultures axenic? Or that was as separate experiment that comes in later?

Reply: We used ampicillin and kanamycin to remove bacteria from Pt1 and PtCr (Lines 404-407), but the bacteria were not completely removed. In order to get strain 'Pt1-sterile', apart from the two antibiotics, ciprofloxacin and streptomycin were also used, and then the diluted culture was coated onto f/2 agar plates with the latter two antibiotics for two weeks (Lines 407-413). The consumption of glucose was not detected in the cultures of 'Pt1-sterile'. 16S rRNA amplicon sequencing showed that abundance of bacterial OTUs is much lower in Pt1-sterile (88) than in Pt1 (380) and PtCr (27884) (a new figure, Fig. 2a).

Lines 125-127: Did you do a co-occurrence analysis? Or this is just based on the presence of these taxa and how that overlaps in the dataset? I would be careful with

co-occurrence or just explain this in more detail in the methods.

Reply: The analysis results of Fig. 4a-b were based on the overlap of *Loktanella*, *Janibacter*, and Diatoms at different sites in the *Tara Oceans* data (Lines 153-155). We have re-evaluated the co-occurrence of diatoms and the two bacteria (*Janibacter* and *Loktanella*) in the revised manuscript (Fig. 4c, Lines 158-160) and provided a detailed description of the analysis method as suggested (Lines 502-507).

Line 193: Did you analyze negative and positive mode features separately or merge them?

Reply: We analyzed all features of positive and negative ion modes separately. The results were merged in Supplementary Table 10 and the source patterns of these features were marked. Meanwhile, the description of the analysis method has been also supplemented (Line 566-569).

Line 290: Consider changing metabolome to metabolomics.

Reply: Thank you. We have made the modification in the revised manuscript (Line 325).

Line 342: It may be nice to have a few sentences to really drive home the broader implications of this work and some of the next steps in the field and lab. For instance, getting to a point of modeling these interactions in the ocean with changing conditions.

Reply: We have expanded the discussion to include broader implications and future directions, together with modeling these interactions (Lines 377-382).

Line 412: It might be nice to add a bit more detail here. Was this based on occurrence of these taxa of interest in the *Tara Oceans* data table? Which specific resource did you use? Link to that resource? Was this from the metagenome or metabarcoding data? Or does macrogenome encompass all these samples?

Reply: The data for *Loktanella*, *Janibacter*, and diatoms were obtained directly from the *Tara Oceans* metagenomic dataset. The file link is: https://www.ebi.ac.uk/biostudies/files/S-BSST297/OM-RGC_v2_taxonomic_profiles.tar.gz. We only used the abundance table belonging to the genus level in this file. We have added corresponding information in the Methods section (Lines 504-506).

Line 432: What about the other 16S sequence data? Has that been archived?

Reply: The sequences of 16S rRNA Sanger sequencing of *J. anophelis* and *L. vestfoldensis* are archived in GenBank under accession numbers PX410800 and PX410801, respectively (Line 448-451). In addition, the raw data of 16S rRNA amplicon sequencing has been deposited in the Sequence Read Archive database under accession number PRJNA1330494 (<https://dataview.ncbi.nlm.nih.gov/object/PRJNA1330494?reviewer=i93stvf24v>

f47fkoaekrs8ogq9) (Line 479-481).

Line 443: Here and elsewhere in the methods, I would consider adding a citation for R packages that you use, if they are available.

Reply: We have added citations for all major R packages (including ggplot2, Fastp, Hisat2, FeatureCount, DESeq2, and ClusterProfiler) used in the Methods section (Lines 492 and 524-534).

Line 455: How much of the supernatant did you collect? Was it roughly the same across samples and replicates?

Reply: For different samples and replicates, we collected 30 mL of supernatant. We have provided the description in the Methods (Line 547).

Line 456: How much volume from each sample for this pooled sample? Did you run the pooled sample once or multiple times on the mass spec?

Reply: We take 200 μ L from each sample to prepare the pooled sample, which is divided into 4 equal parts. The first three pooled samples are used to monitor the precision of the instrument, and the last one is used to monitor the stability of the instrument. We have provided the description in the Methods (Lines 548-550).

Line 469: How much sample volume did you inject? Did you have instrument blanks to account for baseline features in the mobile phase? Just a few more details here would be nice. Also, is this metabolomics data archived somewhere (e.g., MetaboLights)?

Reply: The injection volume for each sample was 2 μ L, and methanol was used as a blank sample to deduct baseline features from the mobile phase. In addition, we have uploaded the metabolome data on Figshare at the following link: <https://figshare.com/s/d84ef77b66aa6082d1a8>. We have provided the detailed descriptions in the Methods section as suggested (Lines 559-561).

Line 472: Citation for MS-DIAL?

Reply: We have added citations for MS-DIAL and metabolomics analysis methods (Lines 566, 576, 578).

Line 479: Did you normalize the feature intensities across samples? For instance, normalizing based on volume or weight. Did you do any other filtering steps, like removing a feature if not found in 2/3 replicates? Or subtracting the intensity from the blanks?

Reply: For metabolomic data, we first used an absolute threshold of peak intensity (Sample/Blank) ≥ 3 for blank deduction, then used the condition of at least 2 feature peaks appearing in repeated samples for feature peak screening, and finally normalized all feature peak data using local weighted regression (lowess) (Lines 566-569).

Line 486: These are more so curiosities or ideas for next steps that may be nice to

consider. It would be helpful to have a host-only treatment as well, so that you could really see what features are specific to the co-culture treatment. Also, I would be interested to run a targeted approach to look at some of these compounds in more detail and/or provide more context to the transcriptome results. These are just cool next steps, that you could mention in the discussion section or expand upon.

Reply: We fully agree this is an excellent idea for future work. We have now mentioned this as an important future direction in the discussion (Lines 377-382).

Figure 3: Can you give more details in the legend about what the circles and their sizes represent. I know it is OTU abundance, but this should be specified in the legend. It would also be nice to explain the insert bar chart. Can you make that bar chart larger? It is difficult to read the axes labels.

Reply: We have moved the insert bar chart in Fig. 4 to Fig. 4b and explained the meaning of the pie chart and its size in the legend. In addition, we have conducted co-occurrence analysis using Tara Oceans data and discovered the correlation between diatoms and *Loktanella* (Fig. 4c), and results of co-culture experiments between *L. vestfoldensis* and two diatoms (*C. muelleri* and *T. pseudonana*) are shown in Fig. 4d,e.

Figure 5: More details in the legend would be nice. Is this showing just the 84 compounds from the MS2 data? This is not all the features, right? I would specify that.

Reply: 84 metabolites from MS2 data were used in Fig. 6. We have provided the details in the figure legend.

Reviewer #2 (Remarks to the Author)

Diatoms are important primary producers, and the interaction between diatoms and their associated bacteria are vital for the marine ecosystem's productivity, nutrient cycling, and overall biogeochemical processes. The authors isolated *Loktanella vestfoldensis* from a culture of the model diatom *Phaeodactylum tricornutum* and found that its interaction with *P. tricornutum* was dependent on the carbon source. *L. vestfoldensis* showed dependence on *P. tricornutum* when CO₂ was the sole carbon source, and growth of the diatom depended on the support of *L. vestfoldensis* for the supply of necessary carbon source when glucose served as the sole carbon source. Genes involved in inorganic carbon concentration, Calvin cycle, and photosynthesis were found to be upregulated in the co-culture of *L. vestfoldensis* and *P. tricornutum*.

Co-culture of *L. vestfoldensis* and *P. tricornutum* transcriptionally upregulated genes involved in the inorganic carbon concentration, Calvin cycle, and photosynthesis, while upper glycolysis remained unaffected. These findings highlight the importance of bacterial-algal interactions in diatom carbon utilization and contribute to marine carbon cycling and redistribution.

The experiments were thoughtfully designed, and the manuscript is well-written. The findings of this study hold significant implications for the field of phytoplankton ecology. However, certain aspects could be refined to better articulate the scientific questions and improve overall clarity for readers.

1. Line 31: "Depletion of inorganic carbon in the surface seawater, compounded by intense phytoplankton photosynthesis, poses constraints to the growth of diatoms." In surface seawater, dissolved inorganic carbon (DIC) primarily exists as bicarbonate (HCO₃⁻), aqueous CO₂, and carbonate (CO₃²⁻). Among these, bicarbonate dominates the

DIC pool due to its high concentration, while dissolved CO₂ remains comparatively low because of its limited solubility under surface ocean conditions. Please rephrase this sentence.

Reply: We have rephrased this sentence as suggested (Lines 33-36).

2. Line 372: Incubation was carried out in vent cap (AC, MC) or plug-seal cap (CL,GC) culture flasks with (MC, GC) or without (AC, CL) the supplement of 3 g/L glucose. Thus, AC stands for atmospheric CO₂ as the sole carbon source, MC for atmospheric CO₂ and glucose as carbon sources, CL for carbon-limited conditions, and GC for glucose as the sole carbon source.

While the abbreviation 'MC' was properly defined in the Methods section, its first occurrence in the Results (Line 84) lacks a parenthetical reminder of its meaning. For optimal readability, we recommend either: reintroducing the abbreviation parenthetically when first used in later sections (e.g.,MC: mixed carbon)

Reply: We have reintroduced the key abbreviations (CL, MC) with their meanings when first used in the Results section as suggested (Line 84 and 92).

3. To clarify the experimental treatments:

Pt1: Received standard antibiotic treatment to partially remove associated bacteria.

Pt1 sterile: Underwent intensive antibiotic treatment to achieve complete bacterial removal.

Pt Cr: Served as an untreated control, maintaining its native bacterial community.

While these protocols were described in the Methods, we suggest explicitly restating this critical distinction in the Results or Discussion to enhance reader comprehension of the experimental design

Reply: We have explained the difference of strains Pt1, PtCr and Pt1-sterile in the Results as suggested (Lines 100-102).

4. I wonder *L. vestfoldensis* is the only bacteria species you isolated from the *P. tricornutum* culture?

Reply: In the revised manuscript, the 16S rRNA amplicon sequencing of the algal cultures without any antibiotics treatment (Fig. 2, Supplementary Fig. 4, Lines 115-123) has been performed, and the results show that both Pt1 and PtCr contain various epiphytic bacteria. We only isolated and purified *L. vestfoldensis* from PtCr and *J. anophelis* from Pt1 after the cultures were treated with ampicillin and kanamycin.

5. Line 397: "To validate the function of *Loktanella vestfoldensis*, the epiphytic bacterium isolated from PtCr cultures, 1 mL of the bacterial culture (OD₆₀₀ ≈ 0.4) was added to the cultures of the 12 *P. tricornutum* strains, and 1 mL of sterile artificial seawater used as the control. Cultivation was performed under GC condition as mentioned above. "

How about the initial cell concentration of *P. tricornutum* in the co-culture?

Reply: We have provided the detail of initial cell density in Methods (Line 454).

Reviewer #3 (Remarks to the Author):

The manuscript addresses the question of species interactions between diatoms and bacteria. The authors picked the model *P. tricornutum* and two co-occurring bacteria from laboratory cultures to investigate as to how inorganic vs organic carbon supply impacts their interactions under light and darkness and in the presence and absence of the bacteria.

Although I think this research is timely, I had some issues with this manuscript. First, I struggled to understand the main results based on reading the abstract. Hence, I think the authors do need to be doing a better job in summarising the main results in a concise and accessible way. Even though I continued reading, I still think the authors should be adding a schematic diagram that illustrates the experimental setup and how the different strains of the diatom have been used. Although the results are interesting and worth following up on, I have methodological and conceptual issues with the presented work.

Reply: We have rewritten the abstract and provided a schematic diagram of the experimental design in Supplementary Fig. 13 as suggested.

Conceptual issues:

A) I think picking *P. tricornutum* as the diatom model was a bad choice considering what the overarching question is. My reasons are as follows:

1) The authors have not provided any evidence beyond culture conditions that *P. tricornutum* is associated with either of the two bacterial species in nature. The lab is a very different environment and this therefore has to be appreciated. To address this concern, I suggest to do co-occurrence analyses to see which are the main diatom species associated with both bacterial species in the oceans opposed to the lab. This can be done by using the TARA Oceans data, and to be honest, this should have been done before setting up any experiments. Some of those data are already available to the authors because they should know from generating figure 3 which are the associated diatom species.

Reply: We have performed co-occurrence analysis on diatoms and the two bacteria (*Loktanella*, and *Janibacter*) using the *Tara* Oceans metagenomic datasets as suggested. Although the datasets do not contain relevant information on *P. tricornutum*, we find a significant correlation between *Loktanella* and ubiquitous diatom genera including *Chaetoceros* and *Thalassiosira* (a new figure Fig. 4c). Therefore, we have co-cultured *Loktanella vestfoldensis* and diatoms (*Chaetoceros muelleri* and *Thalassiosira pseudonana*), and obtained the results (a new figure Fig. 4d,e, Lines 160-163) consistent with those from the model diatom *P. tricornutum*. *L. vestfoldensis* can support the growth of the two diatoms with glucose as the sole carbon source, which provides favorable evidence that the interaction we study in the lab is ecologically relevant.

2) I am sure that *P. tricornutum* does not only associate with two different bacterial species in laboratory cultures. To address this issue, this work needs to explore 16S amplicon sequencing for every diatom strain used in this study to comprehensively assess all interacting bacteria in the laboratory cultures.

Reply: We have performed the 16S rRNA amplicon sequencing on 13 strains of *P. tricornutum* (Pt1-10, PtCr, UTEX640, and Pt1-sterile) together with *Chaetoceros muelleri* and *Thalassiosira pseudonana*. 11 classes, 20 orders, 29 families, and 42 genera are identified, with Alphaproteobacteria and Bacteroidia groups contributing the major OTU abundance. Some marine

bacteria, such as *Caulobacter*, were widely detected in 15 strains of diatoms. However, *Loktanella vestfoldensis* only appears in strain PtCr, implying its unique functionality (Lines 115-123). The 16S rRNA amplicon sequencing data can be accessed through the following link:

<https://dataview.ncbi.nlm.nih.gov/object/PRJNA1330494?reviewer=i93stvf24vf47fkoaekrs8ogq9>.

B) How meaningful are experiments with marine photosynthetic organisms in an environment that does not contain bicarbonate? The latter is the main source of DIC (Dissolved inorganic carbon) in the oceans for many photosynthetic organisms including diatoms (e.g., through using an effective carbon concentrating mechanism). Thus, experiments that do not at least control for the presence of bicarbonate, miss out on ecological relevance and therefore wider significance and applicability of the results.

Reply: We thank the reviewer for this important comment on ecological relevance. We fully acknowledge that our simplified experimental system (atmospheric CO₂ as the sole inorganic carbon source) does not fully recapitulate the natural marine DIC pool (which is bicarbonate-dominated). We have discussed this limitation in our revised manuscript.

We set five different carbon source conditions in our experiments: NC (f/2 with 2mM NaHCO₃⁻, as the control), AC (atmospheric CO₂ as the sole carbon source), MC (atmospheric CO₂ and glucose as carbon sources), CL (carbon-limited conditions), and GC (glucose as the sole carbon source). Addition of 2 mM NaHCO₃ in the medium (NC) did not promote the growth of *P. tricornutum*, indicating inorganic carbon was enough under AC condition. In addition, the growth of *P. tricornutum* was pretty much the same under AC and MC conditions, indicating when DIC was available the addition of glucose did not enhance cell growth. In order to decipher the function of bacteria, we had to grow them under extreme condition such as with organic carbon glucose as the sole carbon source. Results showed that diatoms survived with the carbon support from the bacterium. We have provided a new supplementary figure (Supplementary Figure 12, Lines 421-424).

Methodological issues:

A) Provide evidence of axenic diatom cultures. As no evidence has been provided, I doubt that those cultures have been grown under axenic conditions although the results are different between the axenic vs xenic experiments. But this needs to be clarified, nevertheless.

Reply: Before growth experiments, we performed antibiotic treatments, including standard antibiotic treatment (ampicillin and kanamycin) and intensive antibiotic treatment (ampicillin, kanamycin, ciprofloxacin and streptomycin). After the intensive antibiotic treatment, 'Pt1-sterile' was obtained. Although no bacterial colonies were detected on the LB plate after Pt1-sterile was plated, 16S rRNA amplicon sequencing showed the lowest OTUs abundance (88) in Pt1-sterile (Fig. 2, Lines 115-123). In detail, there is almost only one bacterium, *Caulobacter* (75), which is also present in the other

14 diatom samples.

B) I have not seen any robust evidence that the associated bacteria are symbionts (line 317).

Reply: We agree that the term “symbionts” is not appropriate and we have replaced “symbiotic partnership” and “symbiosis” with “interaction” (Lines 352, 375).

C) No information has been given on the CO₂ concentrations. I have not even seen how CO₂ has been added. Were the cultures bubbled with air? Please provide details and CO₂ concentrations.

Reply: The cultivation was carried out in gas-permeable flasks on an orbital shaker, which allowed CO₂ supply through passive diffusion from the air, rather than bubbling with air or CO₂-enriched air. In the revised manuscript, we measured inorganic carbon concentrations of 3 *P. tricornutum* strains together with *C. muelleri* and *T. pseudonana* under CL, AC, and GC condition using the total organic carbon analyzer (Vario TOC cube, Elementar, Germany). The initial inorganic carbon concentration was about 0.87~1.08 mM, and was 0.45~0.94 mM under AC, 1.12~2.07 mM under GC, 0.10~0.39 mM under CL after 8 days of cultivation. We have provided Supplementary Fig. 2 (Lines 98-99) in the revised manuscript.

D) With referring to B) under conceptual issues, what was the pH of the cultures? Was there a buffer used? The pH has a significant impact on the growth of diatoms and overall metabolism. Thus, pH data need to be provided.

Reply: Normal f/2 medium without buffer was used in our study. We have provided the pH values as suggested in the revised manuscript (Supplementary Fig. 1c, Lines 96-98).

E) Why was no silicate provided? This adds to concerns of wider applicability to the natural system as most diatoms require silicate to grow. *P. tricornutum* appears to be the odd ball out.

Reply: *P. tricornutum* is a model diatom species and has no obligate requirement for silicate for growth. In the revised manuscript, to clarify the impact of silicate deficiency on the overall experimental results, we have used seawater supplemented with silicate and f/2 nutrients to re-cultivate 13 strains of *P. tricornutum* (Pt1-10, PtCr, UTEX640 and Pt1-sterile). The results show that the growth and glucose consumption of *P. tricornutum* are consistent with the previous results (Supplementary Fig. 1a,b, Lines 90-91), indicating that silicate has no significant effect.

REVIEWERS' COMMENTS

Reviewer #1 (Remarks to the Author):

The authors have implemented my suggested edits and I have no further concerns with the manuscript at this time. Thanks for all of your hard work on improving the paper!

Reply: We are greatly encouraged by the reviewer's kind words and appreciate the positive assessment of our work.

Reviewer #2 (Remarks to the Author):

Thank you for revising the manuscript in response to the reviewers' comments and suggestions. From my perspective, you have fully addressed the questions I raised in the first round of review.

Reply: We are grateful to the reviewer for the constructive suggestions which help to improve our paper.

Reviewer #3 (Remarks to the Author):

The revised version of this manuscript has addressed all of my comments. The additional experiments and data have made the results more robust and relevant for a broader community of scientists. Weaknesses have been addressed appropriately. Congrats to this thorough revision! I have only some minor suggestions to improve the manuscript before it can be accepted. I do not need to see the next version anymore.

Reply: We appreciate all the comments from the reviewer and feel greatly encouraged by the reviewer's kind and positive assessment of our work.

- Figures 1-4 miss to mention the number of replicates (N=?) in the figure legends. Also, actual exact values for "p" should be provided and the test that has been used to calculate them.

Reply: We have added the number of replicates, p-values, and other statistical information in the legend of Fig. 1-4.

- Figure 2 has too much redundant information. I think the level of organisation that matters the most considering the rest of the results is the genus level. Hence, move panels a, b, and c to the supplement.

Reply: Following your suggestion, we have modified Fig. 2 by retaining only Fig. 2d, and have moved Fig. 2a-c to Supplementary Fig. 4.

- I don't think figure 7 is necessary. The information presented in this figure is simplistic and has already been well described and discussed in the text of the manuscript. Either delete it or move it to the supplement. Another reason: similar cartoons have been published before. What maybe would make sense is to extend existing concepts of diatom-bacteria interactions with the new data presented in this manuscript.

Reply: We have moved Fig. 7 to Supplementary Fig. 12 as suggested.